# *Acanthamoeba castellanii* as a Screening Tool for *Mycobacterium avium* Subspecies *paratuberculosis* Virulence Factors with Relevance in Macrophage Infection

**DOI:** 10.3390/microorganisms8101571

**Published:** 2020-10-13

**Authors:** Ida L. Phillips, Jamie L. Everman, Luiz E. Bermudez, Lia Danelishvili

**Affiliations:** 1Department of Biomedical Sciences, Carlson College of Veterinary Medicine, Oregon State University, Corvallis, OR 97331, USA; ida.phillips@oregonstate.edu (I.L.P.); Luiz.Bermudez@oregonstate.edu (L.E.B.); 2Center for Genes, Environment, and Health, National Jewish Health, Denver, CO 80206, USA; evermanj@njhealth.org; 3Department of Microbiology, College of Sciences, Oregon State University, Corvallis, OR 97331, USA

**Keywords:** *M. avium* subsp. *paratuberculosis*, amoeba, macrophage, virulence factors

## Abstract

The high prevalence of Johne’s disease has driven a continuous effort to more readily understand the pathogenesis of the etiological causative bacterium, *Mycobacterium avium* subsp. *paratuberculosis* (MAP), and to develop effective preventative measures for infection spread. In this study, we aimed to create an in vivo MAP infection model employing an environmental protozoan host and used it as a tool for selection of bacterial virulence determinants potentially contributing to MAP survival in mammalian host macrophages. We utilized *Acanthamoeba castellanii* (amoeba) to explore metabolic consequences of the MAP-host interaction and established a correlation between metabolic changes of this phagocytic host and MAP virulence. Using the library of gene knockout mutants, we identified MAP clones that can either enhance or inhibit amoeba metabolism and we discovered that, for most part, it mirrors the pattern of MAP attenuation or survival during infection of macrophages. It was found that MAP mutants that induced an increase in amoeba metabolism were defective in intracellular growth in macrophages. However, MAP clones that exhibited low metabolic alteration in amoeba were able to survive at a greater rate within mammalian cells, highlighting importance of both category of genes in bacterial pathogenesis. Sequencing of MAP mutants has identified several virulence factors previously shown to have a biological relevance in mycobacterial survival and intracellular growth in phagocytic cells. In addition, we uncovered new genetic determinants potentially contributing to MAP pathogenicity. Results of this study support the use of the amoeba model system as a quick initial screening tool for selection of virulence factors of extremely slow-grower MAP that is challenging to study.

## 1. Introduction

*Mycobacterium avium* subsp. *paratuberculosis* (MAP) is an etiologic agent of a chronic and highly contagious granulomatous enteritis known as Johne’s disease. The MAP infection primarily effects the domestic and wild ruminants worldwide. The infection generally occurs early in neonatal calves that do not develop disease for several years and later serve as a carrier for MAP infection among healthy herds. In the United States, roughly 68–91% of dairy herds are infected with MAP, causing the biggest losses in the cattle dairy industry [1,2]. Johne’s disease does not currently have a cure and while antibiotic therapy can slow the appearance of disease, historically, it has been shown to be ineffective and uneconomical. In order to find a solution for control of MAP infection and develop effective treatment, management, and diagnostic strategies, it is essential to understand molecular mechanisms of MAP pathogenesis.

Johne’s disease progresses slowly, following MAP uptake by macrophages in the intestinal tract. Macrophages are preferred host cells for MAP infection, and the ability of pathogenic mycobacteria to resist to the killing mechanisms of this professional phagocyte cell is essential for bacterial survival, persistence, and spread within the animal host. Similar to other mycobacterial species, after uptake by macrophages, MAP hijacks or inhibits cellular mechanisms and prevent phagosome-lysosme fusion [3,4,5]. By preventing the host phagocyte tactics, normally employed to clear intracellular pathogens, MAP successfully establishes a niche and thrives within the vacuoles, forming granulomatous lesions within the intestinal tissue. Bacterial dissemination ultimately results in a severe wasting and death of the animal [6].

Intracellular bacteria residing within the phagocytic vacuole rely on nutrients found within the host over the course of infection, and pathogens develop mechanisms to obtain the required nutrients during infection [7]. It is well known that, in the environment, mycobacteria can evolve survival strategies from protozoan hosts. Early studies revealed that *Mycobacterium avium* species can replicate in amoeba *Dictyostelium discoideum* [8,9] and the phenotype that mycobacteria develop within *Acanthamoeba castellanii* is considerably more virulent than the one expressed in bacteria of culture medium [9,10,11]. For opportunistic pathogens that can be acquired from the environment, these virulence traits likely evolve to increase fitness of the pathogen. The fact is that MAP resides in the environment after being shed in the feces from infected animals and can interact with a variety of species of protozoa in the soil. While MAP has been frequently isolated from amoeba [12], free-living amoeba has been frequently isolated from habitats common to mycobacteria [13,14], supporting an “endosymbiotic” relationship between mycobacteria and free-living protozoan host [15,16]. In addition, while mycobacteria have evolved mechanisms to survive within amoeba, amoeba would possibly protect phagocytized mycobacteria from adverse environmental insults, including extreme temperature, drought, and diverse biocide attacks via cyst formation [17].

In recent years, amoeba has become an important model organism to study pathogenicity mechanisms of *Francisella, Legionella, Salmonella*, or *Mycobacterium* [18]. Amoeba and professional phagocytes, such as macrophages, share highly conserved microbicidal mechanisms and machineries including acidification and maturation of phagosomes, lysosomal degradation, production of reactive oxygen species (ROS), and immune defenses influenced by micronutrients [19,20,21]. In addition, amoeba possesses genes that are involved in bacteria recognition, uptake by phagocytosis mechanism, cytoskeleton rearrangement and phagosome formation, and many fundamental signaling pathways that are common with macrophages. Results obtained by our group demonstrate high similarities in mycobacterial genes expression during amoeba and macrophage infection [22] and provide the solid evidence that amoeba can serve as a useful model for intracellular MAP infection.

In current study, we used *Acanthamoeba castellanii* as a model system to determine the influence that MAP infection has on amoeba metabolic activity. We detail how metabolic changes of amoeba directly mirrors the pattern of MAP survival and intracellular burden in host macrophages. In addition, by screening a transposon bank of MAP mutants, we discovered virulence factors that are used by the pathogen to overcome host defenses. These data aid in understanding some mechanisms that drive the interaction between MAP and phagocytic cells.

## 2. Materials and Methods

### 2.1. Bacterial Strains and Cultures

*Mycobacterium avium* subsp. *paratuberculosis* strain K10 (MAP) purchased from American Type Culture Collection (ATCC, BAA-968) was used for all experimental procedures. Bacteria were cultured on 7H10 Middlebrook agar medium enriched with casein hydrolysate (1 g/L; BD), 10% (vol/vol) oleic acid, albumin, dextrose, and catalase (OADC; Hardy Diagnostics; Santa Maria, CA, USA), and ferric mycobactin J (2 mg/L; Allied Monitor, Fayette, MO, USA) and incubated at 37 °C for 3–4 weeks. Inoculum was prepared in phosphate buffered saline (PBS), passed through a 22-gauge needle, and suspension was allowed to settle for 10 min. The top 2/3 was moved to a new polystyrene tube and the mixture was passage through a 22-gauge needle again. The inoculum was adjusted to match McFarland standard #1 and serial dilutions were plated for colony forming unit counts (CFUs) for a calculation of exact bacterial concentration.

### 2.2. Amoeba Culture

*Acanthamoeba castellanii* was purchased from ATCC (ATCC 30234). The amoeba was cultured in the T-25 flask in 712 Peptone-Yeast-Glucose (PYG) media consisting of 10 g proteose peptone, 0.5 g yeast extract, 4 mM magnesium sulfate heptahydrate, 400 μM calcium chloride, 3.4 mM calcium chloride, 50 μM ferric ammonium sulfate hexahydrate, 2.5 mM disodium phosphate heptahydrate, 2.5 mM potassium phosphate, 0.1 M glucose, and pH 6.5. Axenic cultures were placed in dark at 25 °C. For experiments, amoeba was washed from the flask surface in fresh media and seeded to appropriate density as per experimental protocol.

### 2.3. Mammalian Cell Culture

RAW 264.7 murine macrophage were purchased from ATCC (TIB-71) and grown in Dulbecco’s Modified Eagle’s Medium (DMEM) supplemented with 10% heat-inactivated fetal bovine serum (FBS; Gemini Bio-Products; West Sacramento, CA, USA) at 37 °C in 5% CO_2_ incubator. Cells were allowed to grow 80–100% confluence for infection assays.

### 2.4. Construction of MAP Transposon Library

The MycomarT7 (Mmt7) phagemid containing temperature sensitive transposon was provided by Dr. Eric Rubin at the Harvard T.H. Chan School of Public Health, Boston, MA. Mmt7 was propagated in the fast-growing *Mycobacterium smegmatis* mc^2^155 strain, and the phage stock concentration was determined using titration method [23]. MAP was transduced as previously described [24] with slight modifications. MAP cultures grown in the 7H9 Middlebrook broth medium supplemented with 10% OADC and 0.1% Tween-80 from the mid-log phase were centrifuged at 3500 rpm for 20 min and washed with MP buffer (150 mM NaCl, 50 mM [pH 7.5] Tris-HCl, 10 mM Mg_2_SO_4_, 2 mM CaCl_2_) two times. Bacteria were resuspended in MP buffer and adjusted to McFarland standard #3 (9 × 10^8^ CFU/mL). A 5-mL volume of this suspension was infected with Mmt7 at a multiplicity of infection (MOI) of 2 phages to 1 bacterium. MAP was transduced for 4 h with gentle agitation at 37 °C. To obtain individual transposon mutants, aliquots were plated on 7H10 agar containing 200 µg/mL kanamycin. The kanamycin-resistant colonies were randomly selected and screened for the presence of Mmt7 with PCR. Next, 27 96-well plates were generated by placing MAP individual mutant colonies in each well and grown in 300 uL of Middlebrook 7H9 broth with OADC (10% *v*/*v*), mycobactin J (2 mg/mL), and 200 µg/mL kanamycin sulfate at 37 °C for 45 days.

### 2.5. Transposon Mutant Sequencing

MAP mutants were lysed in the deionized water with mechanical disruption method using 0.2 mL of 0.1-mm-diameter glass beads. Samples were cleared by centrifugation at 10,000× *g* for 5 min and DNA was purified using a DNA Clean and Concentrate kit (Zymo Research, Irvine, CA, USA) following the manufacturer’s protocol. The transposon insertion site was identified using the previously reported ligation-mediated polymerase chain reaction (LM-PCR) technique that is comprehensively described in [24]. The final PCR products were separated on the agarose gel, and purified DNA fragments sequenced at the Center for Genome Research and Biocomputing, Oregon State University.

### 2.6. Visualization of Intracellular MAP within Amoeba

Amoeba cultures seeded onto glass chamber slides at 80–100% confluence in 712 PYG media were infected with MAP at MOI of 10 and incubated for 1 h at 25 °C. Slides were washed three times with PBS to remove extracellular bacteria, heat fixed, and stained using Kinyoun acid-fast protocol for visualization of intracellular mycobacteria on a DM4000B Leica microscope under bright field. In addition, because the carbol-fuchsin stain (part of the Kinyoun acid-fast kit) emits red autofluorescence, we labeled MAP with carbol-fuchsin for 1 h. Bacteria were washed three times with PBS and infect amoeba. After 1 h infection, amoeba was washed with PBS, fixed with 4% paraformaldehyde for 20 min and images were captured on a Leica DM4000B fluorescent microscope (Leica) with QCapture Pro7 software (QImaging; Surrey, BC, Canada) and using the brightfield and Texas Red filters.

### 2.7. MAP Survival during Amoeba Infection and Amoeba Viability

Amoeba was cultured in 24-well plate overnight at 25 °C and, the next day, confluent monolayers were infected with MAP at MOI of 10. The infection was synchronized by centrifuging plates at 225× *g* for 5 min and incubated for 2 h at 25 °C in the dark. The wells were gently washed with the Page’s amoeba saline (ATCC) after 2 h of post-infection (0 time point) to remove extracellular bacteria. At 0, 1, 2, 3, 5, 7, 10-, and 15-days post-infection, both plate-adhered amoeba and non-adhered amoeba from supernatant were lysed in 0.5% sodium dodecyl sulfate, passed through a 28-gauge insulin needle, and pelleted at 2000× *g* for 10 min. The serial dilutions of bacterial pellets were plated to quantify viable intracellular bacteria recovered from amoeba. In addition, duplicate wells were suspended in the Page’s amoeba saline, stained with Trypan Blue (0.4% solution), and quantified for cell viability over the same time course.

### 2.8. Establishing Amoeba Metabolic Activity

The 96-well tissue-culture plates were seeded with amoeba overnight and were infected with wide-range of bacterial MOIs (100, 50, 10, 1). The only 712 PYG media wells served as a negative control. Infections were synchronized by centrifugation at 225× *g* at 25 °C for 5 min. Plates were incubated for 2 h at 25 °C in the dark, at which time wells were washed 2 times with the Page’s Amoeba Saline. Fresh 712 PYG media supplemented with 10% (*v*/*v*) AlamarBlue (Life Technologies, Carlsbad, CA, USA) was added to each well. Fluorescent readings were measured at 530 nm/590 nm (excitation/emission) every hour for 24 h using a Tecan F200 plate reader (Tecan Group Ltd., Männedorf, Switzerland). For long-term experiments, amoeba was infected at established MOI of 10:1 as described above, and fluorescent readings of alamarBlue dye (10% *v*/*v*) were taken at time 0, 2, and every 12 h until 180 h post-infection. The 712 PYG media, live MAP alone, heat-killed MAP, and latex beads were inoculated into wells in the absence of amoeba and treated in identical manner and served as controls.

### 2.9. Metabolic Screening of MAP Mutant Library Using Alamarblue

*Acanthamoeba castellanii* were seeded in 96-well plates with ≥80% and allowed to rest for 2 h in 712 PYG. The amoeba monolayers were infected with the wild-type MAP or transposons mutants at a MOI of 10:1. The uninfected and the wild-type infected amoeba served as negative and positive controls. The infections were synchronized by centrifugation at 250 rpm at 25 °C for 5 min. After, 2 h incubation at 25 °C in the dark, wells were washed with the Page’s amoeba saline and 300 uL of 712 PYG media supplemented with 10% (*v*/*v*) AlamarBlue was added to each well. Fluorescent measurements were taken at 530 nm/590 nm at 24 h of post-infection using a Tecan F200 plate reader. In total ~2500 individual MAP transposon mutants were tested for the ability to alter the amoeba metabolism (stimulation or inhibition) and bacterial clones that triggered 2-fold or more changes compared with the wild-type infection were selected for sequencing.

### 2.10. Macrophage Invasion and Survival Assays

The Raw 264.7 macrophages (10^6^) were seeded in 24-well plates overnight in DMEM and infected with MAP or transposons mutants at a MOI of 10 bacteria: 1 cell. Plates were synchronized at 250 rpm for 5 min and incubated at 37 °C. After 2 h infection, supernatants were removed, and the wells were washed three times with Hank’s buffered salt solution (HBSS) or PBS. In addition, DMEM media supplemented with amikacin (200 μg/mL) was added for additional 2 h to kill any remaining extracellular bacteria. Macrophage monolayers were lysed with 500 μL of 1% Triton X-100 in HBSS at 24 h and 96 h time points, serially diluted, and then plated for CFU determination on 7H10 agar plates.

### 2.11. Complementation of MAP Mutants 

The selected MAP gene knockout mutants (3D3, 7C1, 15G2, 18D6) were complemented with the functional genes using the chromosome integration pMV306 plasmid containing apramycin resistant marker [25]. Primers are listed in the Table 1. Electrocompetent MAP mutant cells were prepared by washing plate-grown bacteria four times and centrifuged at 2000× *g* and 4 °C for 25 min with a chilled solution of 10% glycerol and 0.1% Tween 80. Resulting plasmid constructs verified by double restriction digestion were electroporated into the respective MAP mutant using the following setting: 2500 V, 1000 Ω, and 25 µF and 0.2 cm cuvette. Bacteria were cultured in 7H9 broth supplemented with OADC (10% *v*/*v*) and mycobactin J (2 mg/mL) at 37 °C for 24 h, followed by plating on 7H10 plates containing 200 µg/mL apramycin for 10–12 weeks. To confirm positive transformation, MAP colonies were PCR screened using gene specific primers. Next, to determine if the phenotype has been restored, complemented clones were tested for infection and survival in macrophages.

### 2.12. Statistics

Statistical comparisons between experimental and control groups were made using a two-tailed unpaired *t* test. Detailed information about data presentation and *p* value definitions are listed in figure legends. Graphical outputs were created in GraphPad Prism software.

## 3. Results

### 3.1. MAP is Readily Taken up by and Replicate within Acanthamoeba castellanii

In order to utilize amoeba as a proper phagocytic host and a model for the metabolic interaction that occur during MAP infection, we first established the parameters of amoeba infection. Figure 1 shows that amoeba readily uptake bacteria after 1 h of infection and MAP can be seen in large number within the intracellular vacuoles. Using Kinyoun stain, the intracellular MAP is identified as acid-fast positive pink bacilli (Figure 1A) and, using carbol-fuchsin labeled bacteria, red bacilli are observed under fluorescent microscope (Figure 1B).

To understand the long-term outcome of MAP infection on amoeba and determine the survival dynamics of intracellular MAP, *A. castellanii* cultures were infected and both extra- and intracellular bacterial viability was monitored over 15 days (Figure 2A). Results reveal that the amount of MAP quantified after 1 h infection is nearly 100% of the initial inoculum, which indicates that amoeba readily phagocytose the majority of MAP upon infection. Within 24 h of infection the amoeba eliminates almost 50% of ingested bacteria and maintain similar intracellular bacterial burden up to three days. At day 5 of post-infection, the intracellular MAP starts to replicate followed with increased number in viable bacteria, and by day 7, roughly reached the number of original intracellular bacteria that was ingested during initial infection. At 7, 10, and 15 days of post-infection, amoeba number exceed the space constraint of the plate and start to detach into the media (observed by microscopy), where they remained viable. In addition, the extracellular MAP is observed in the media at 15 days of post-infection. While amoeba can suppress MAP growth during first three days of infection, it later appears to serve as a host for the pathogen replication (5 to 10 days). MAP infection, however, does not have a detrimental effect on amoeba viability (Figure 2B). The number and growth rate of infected amoebae in experimental wells are seen to be similar to the uninfected control over the same time course. Our data indicate that *A. castellanii* has the ability to ingest the majority of MAP, and we can establish the temporal pattern of the intracellular and extracellular MAP viability as well as amoeba viability over the course of a 15-day infection.

### 3.2. Early Metabolic Response of Amoeba to MAP Infection

To determine the overall metabolic activity that amoeba undertake during MAP infection, we employed alamarBlue dye as an indicator of oxidative respiration. The fluorescence values obtained from MAP infected amoeba indicate that the chief changes in amoeba metabolism was seen within the first 24 h of infection compared with the uninfected control (Figure 3), and the outcome reflects the fact that, in initial 24 h, amoeba is able to eliminate roughly 50% of ingested MAP (Figure 2A). The stimulation of amoeba metabolism was dose dependent (data not shown), and the infection with MOI of 10 resulted in a significant increase in metabolism (*p* < 0.0001) when compared with uninfected control groups (Figure 3). Since bacterial replication within infected host can also stimulate a shift in AlamarBlue fluorescence, we included MAP control and investigated if bacterial growth and metabolic activity could be a contributing factor to physiological changes observed in amoeba. In the absence of amoeba, MAP alone results in a minimal fluorescence over 24 h time of the assay. In addition, to establish whether the stimulation of amoeba metabolism was an active process triggered by live MAP, we infected amoeba with the heat-killed bacteria and determined that neither the heat-killed bacteria alone nor the heat-killed MAP infected amoeba exhibits metabolic changes that are observed in the live MAP infection group (Figure 3). To confirm that the metabolic stimulation observed in this assay was due specifically to the ingestion of the pathogen, and not a bystander effect that occurs during the process of phagocytosis, an assay was conducted with latex beads. Upon uptake of latex beads, there is negligible stimulation of metabolic activity of amoeba recorded over the course of the assay (Figure 3). Altogether, our data indicate that amoeba metabolism is highly stimulated by the ingestion of live MAP in a dose- and viability-dependent manner. Amoeba metabolism is dramatically stimulated in response to MAP infection within the first 24 h of infection, coinciding with the decrease in the intracellular burden of bacteria over the same time frame. However, as infection progresses, MAP begin to replicate and grow within the intracellular environment leading to eventual escape into the extracellular environment.

### 3.3. Late Metabolic Response of Amoeba to MAP Infection

To determine the metabolic state of amoeba during later stages of MAP infection, the alamarBlue assay was conducted over eight days (Figure 4). Consistent with the data shown in Figure 3, within the first 24 h the metabolic activity of MAP-infected amoeba is dramatically increased (Figure 4a). Between one and three days of post-infection, the metabolism of infected amoeba remains higher than uninfected amoeba cultures, though a much lower alteration when compared to the initial infection (Figure 4b–e). These data combined with the observation that there is no increase in intracellular MAP number seen in the same (1–3 day) time frame (Figure 2A) suggest that the higher metabolic level of the host is able to limit bacterial replication, and possibly MAP has not yet adjusted to an intracellular environment at the early stages of infection. As intracellular MAP begins to replicate (Figure 2A), the metabolic activity of amoeba starts to increase up to five days post-infection (Figure 4g). However, as MAP begins to replicate between 7 and 10 days, this also leads to bacterial escape from the infected phagocytes. At this point, the stimulation of metabolic activity is no longer seen when compared with uninfected amoeba (Figure 4h–j). Together, our data indicate that metabolic changes of amoeba has a direct relationship with the intracellular burden of MAP.

### 3.4. Stimulation of Amoeba Metabolism by MAP Gene Knockout Mutants

Because the infection with live MAP triggers significant increase in amoeba metabolism, we hypothesized that bacterial virulence determinants could be the contributing factors for metabolic stimulation of the phagocytic host. To identify MAP genes associated with this phenotype, a transposon gene knockout library was constructed and, using the alamarBlue assay, screened for significant changes in amoeba metabolism during 24 h of MAP infection (Figure 5). Each library plate of the 96-well plate contained the wild-type MAP infected amoeba and served as an internal control for amoeba metabolic stimulation and for variability between array plates. The bacterial density of each culture plate was measured and adjusted to ensure similar MIC rates in all wells. Of the ~2500 mutants screened, the highest and the lowest metabolic curves were identified and corresponding MAP mutants with ≤ and ≥2-fold alteration over the wild-type control were selected (Figure 5A). MAP mutants were retested for amoeba metabolic assays in three independent experiments and number of clones with 2-fold greater or lower changes over the wild-type control were found to be 27 and 18, respectively (Figure 5B). Using the LM-PCR, we sequenced 35 mutants that are listed in the Table 2.

### 3.5. Validation of Metabolic Mutants in Macrophage Assay 

To determine if distinct metabolic changes observed in amoeba during MAP gene knockout mutant infection had any relationship with the virulence mechanism of bacteria, we tested sequenced bacterial clones for survival phenotype within the mammalian host cells. Using murine macrophages Raw 264.7, mutants were assayed for the ability to invade phagocytic cells (Figure 6A). With the exception of few mutants, the percentage of uptake for each clone was relatively consistent and was not significantly different from the wild-type MAP control. To establish if gene inactivation by transposon made mutants functionally deficient and attenuated in intracellular growth within phagocytic cells, bacterial CFUs were quantified in Raw 264.7 macrophages at 24 h and 96 h post-infection (Figure 6B,C). Bacterial number was normalized to their respective uptake values to account for differences in initial MAP number that entered cells. It has been observed that the majority of MAP clones (except in few exceptions) with low stimulatory effect on amoeba metabolism exhibited equivalent or slightly increased intracellular survival when compared with the wild-type MAP infection at 24 h post-infection (Figure 6B), and differences in survival rate were apparent at 96 h post-infection (Figure 6C). Alternatively, mutants that triggered high metabolic changes in amoeba were found to have a diminished survival rate within Raw 264.7 cells, which was significantly notable at 96 h post-infection (Figure 6C). In few exceptions, some high metabolism MAP mutants were able to resume in growth and maintain a steady rate of infection similar to the wild-type MAP strain.

### 3.6. The Complementation Restores the Attenuation Phenotype in Mutants

In order to validate that phenotypic changes resulting in low MAP survival rate were directly linked to genes knocked out by Mmt7 transposon, we selected four attenuated mutants displaying either low (7C1, 18F6) or high (3D3, 15G2) metabolic stimulation in amoeba, and complemented with functional gene using mycobacterial chromosome integration plasmid pMV306 with apramycin resistant marker.

MAP_0949 is an uncharacterized protein that contains the EAL domain/diguanylate cyclase domain (e-value 5.73925e-99), which is found in diverse bacterial signaling proteins associated with virulence. This group of proteins stimulate degradation of a second messenger cyclic di-GMP, and also can express the diguanylate phosphodiesterase function. The *MAP_3893c* gene encodes the serine/threonine kinase (PknG) protein involved in signal transduction via phosphorylation and it also regulate amino-acid uptake and stationary phase metabolism. MAP_2291 (*glbO*) is described as a truncated hemoglobin group 2 (O) protein that is 86% identical to *Mycobacterium tuberculosis* hemoglobin *glbO* and the *MAP_3634* gene encodes a hypothetical protein containing the IgD-like repeat domain of mycobacterial L, D transpeptidases associated with virulence, and amoxicillin resistance in *M. tuberculosis*.

The complemented clones were first tested for growth in 7H9 broth medium and were found to have similar replication rates as the wild-type MAP (data not shown). The macrophage infection assay presented in the Figure 7 demonstrates that the complementation of selected mutants with the functional genes fully restored the attenuation phenotype of MAP mutants.

## 4. Discussion

Intracellular pathogens have evolved strategies to exploit the host to sustain their survival and replication. Upon entry into macrophages, mycobacteria are sequestered into vacuole compartments where the pathogen establishes distinct protected niche and lifestyle [26]. During the host–pathogen encounter, the intracellular environment restricts mycobacterial replication through nutrient deprivation and directly influence its metabolic networks and virulence [27]. While bacteria activate innate cellular defenses meant to eliminate the infection, pathogens use immune cues to influence gene expression, promoting their fitness in the host microenvironment [28].

Multiple cell types ranging from mammalian macrophages to environmental amoeba can serve as a host to mycobacterial infections. Eukaryotic organisms like amoeba have long been noted to interact with mycobacterial species in the environment and, therefore, are part of the mycobacterial evolution [29]. It has been also demonstrated that intra-amoeba growth can increase bacterial virulence, and some common mechanisms have been implicated with the pathogen’s survival in amoebae as well as in macrophages [19,30]. For example, *Mycobacterium avium* subsp. *hominissuis* (MAH) infection of *A. castellanii* can trigger expression of virulence genes that are also associated with MAH pathogenicity in human macrophages [22]. MAH has been characterized to possess increased virulence when isolated from amoeba [10] and express similar phenotype after infection of mammalian macrophages [31,32]. Likewise, amoeba has been shown to host MAP and serve as a potential reservoir for the long-term bacterial survival in the soil of farm pastures [12] and expressing enhanced resistance to common disinfectant treatments including chlorine and antibiotics [33,34].

Amoeba and macrophages are highly alike in many aspects as they share genes and mechanism for bacterial recognition, uptake, phagocytosis, phagosome formation, and signaling pathways, including interaction with mycobacteria [34]. The comparable characteristics and traits expressed by intracellular MAP during infection of both phagocytes suggest that amoeba may serve as a reliable model to investigate the host-pathogen interaction and the metabolic impact on the host.

The metabolic adaptations by many pathogenic bacteria during intracellular growth are common, and it allows invading organism to profit from host nutrients and other metabolites that otherwise can be toxic. For example, *Salmonella typhimurium* infection of the gut elicits a strong inflammatory response characterized by oxidative stress mechanisms and leads to the production of the sulfur compound tetrathionate, which is a toxic or unusable substrate for most of the microbiota. However, *Salmonella* possess the operon of *ttrABC* genes, involved in the hydrogen sulfide metabolic pathway, enabling bacteria to utilize tetrathionate as an electron acceptor [35,36]. Likewise, *Shigella* is capable of capturing host cell waste products, particularly pyruvate as an energy source, while causing no alterations in the host metabolic activity and maintaining normal fluxes through glycolytic pathways in host cells [37]. *Mycobacterium tuberculosis* also has an ability to break down host lipids and cholesterol, detoxify the breakdown components, and utilize the resulting molecules for the acquisition of carbon during intracellular infection [38].

Recent studies have defined bacterial virulence genes that aid the pathogen not only in evading the host immune mechanisms, but, at the same time, triggering the increased nutrient availability and allowing the pathogen to modify or expand its nutrient requirements [39]. These mechanisms have been termed as ‘nutritional virulence’ and augment bacterial proliferation within the intracellular environment [40,41]. Thus, the nutrient limitation by the host and nutrient acquisition by pathogenic bacteria are crucial processes contributing to the pathogenesis of infectious diseases. Multiple pathogens utilize nutritional virulence as a survival strategy, for example, *Chlamydia* inclusions acquire nutrients from rerouted and fragmented intracellular vesicles [42,43], while *Legionella pnemophila* can trigger proteosome degradation of host proteins in both mammalian cells and environmental amoeba to increase amount of amino acids for nutrients and virulence as well [44].

It has been verified through transcriptomics that MAP undergoes significant genetic modifications within the host intestine [45]. The host environment also influences MAP proteome makeover during in vivo infection, enhancing adaptation pathways [46]. It is, however, unknown if MAP can stimulate the metabolic state of host phagocytes for its own benefit and whether some of genetic factors contributing to MAP pathogenicity could be also related to the host metabolic remodeling. Our study identified the metabolic status of environmental phagocyte during MAP infection and established a common set of MAP genes that are linked to metabolic changes of the protozoan host and to intracellular growth in the mammalian host. We found that MAP infection increases oxidative metabolism in amoeba and, using the library of gene knockout mutants, two bacterial subgroups were identified that could either promote or inhibit metabolic activity in amoeba. The follow-up survival studies performed in Raw 264.7 cells demonstrate that MAP growth of amoeba high metabolic mutants (with few exceptions) was adversely affected during macrophage infection. On the other hand, mutants that did not require increase in amoeba metabolism resulted in enhanced bacterial survival in macrophages.

To confirm if intracellular growth defect in MAP was primarily associated with the gene disrupted by the transposon, we generated complemented clones for macrophage attenuated MAP mutants that exhibited either low-metabolic (7C1, 18F6) or high-metabolic (3D3, 15G2) activity in amoeba. It was found that the functional repair of selected *MAP_0949* and *MAP_ 2291* genes of low metabolism and *MAP_3634* and *MAP_3893c* genes of high metabolism has restored survival phenotype in MAP mutants, and intracellular growth observed from complemented clones were similar to the one seen for the wild-type control growth in Raw 264.7 macrophages.

The *MAP_3893c* gene encodes the serine/threonine-protein kinase G (PknG) protein, which is a well characterized virulence factor of *Mycobacterium tuberculosis.* The targeted knock-out of the *pknG* gene significantly diminishes *M. tuberculosis* survival in vitro and in vivo [47,48], suggesting *pknG* importance for pathogenicity in the eukaryotic host [49]. PknG also contributes to biofilm development and intrinsic antibiotic resistance [50,51]. The serine/threonine-protein kinase G is secreted in the cytoplasm of infected macrophages and translocated to the Golgi complex, where it interacts and blocks the recruitment of active Rab7l1-GTP to pathogen-containing phagosomes, and, subsequently, inhibits the phago-lysosome fusion [52]. The *pknG* knockout mutant, however, is rapidly transferred to lysosomes and killed. PknG is critical for the development of a non-replicating persistence phenotype of *M. tuberculosis* and for the metabolic adaptation under the latency-like hypoxia condition [53]. In addition, PknG has been demonstrated to contribute to stable granuloma formation in the guinea pig model [53]. Moreover, PknG senses amino acid availability and the PknG-GarA signaling pathway controls balanced nutrient utilization and nutritional adaptation to virulence [54]. These studies provide solid evidence on PknG essential role in modulating cellular metabolism for mycobacterial adaptations and survival in the host milieu. Taken together, the indispensable role of PknG in mycobacterial virulence makes this factor an attractive drug target [55]. Although the molecular mechanisms of PknG action has not been described in MAP, due to the fact that this protein has 85% identity to *M. tuberculosis* PknG protein conserved with all functional domains, we can assume that most likely it has similar functions in MAP pathogenicity as well; although, this needs to be experimentally validated.

MAP_0949 is a hypothetical protein that contains the lipoprotein signal peptide with a cleavage site between 26 and 27 amino acids. It is predicted to be cleaved by Signal Peptidase II (*Lsp*) and transported by the Sec translocon (Sec/SPII with a likelihood of 0.058). MAP_0949 protein consists of three domains of EAL, GGDEF, and GAF. EAL domain is found in diverse signaling proteins in bacteria stimulating degradation of a second messenger cyclic di-GMP (c-di-GMP) and, together with the GGDEF domain, it has been proposed to be involved in regulation of bacterial cell surface adhesion [56]. While the EAL domain has been implicated in diguanylate phosphodiesterase function, the GGDEF domain containing proteins have been shown to modulate cyclic diguanosine monophosphate turnover and phosphodiesterase activity [57]. GAF domains, on the other hand, are nucleotide-specific cAMP- and cGMP-regulating domains with wide-range cellular roles [57]. The c-di-GMP and cAMP signaling molecules are second messengers that respond to and control expression of a variety of environmental and quorum sensing signals that bacteria cannot directly internalize, and support microbial transition between motility and surface-associated sessility [58], including biofilm formation and biofilm-associated motilities in opportunistic human pathogens such as Pseudomonas aeruginosa, *Escherichia coli*, *Staphylococcus aureus,* and *Salmonella typhimurium* [56,59,60]. The cyclic di-GMP signaling system regulate several key virulence processes required for bacterial adaptation during the host infection and the evasion of the host immune system, and has been implicated in pathogenesis of *Vibrio cholerae*, *Borrelia burgdorferi*, *Pseudomonas aeruginosa, Yersinia pestis, Clostridium perfringens,* and *M. tuberculosis* [61,62,63,64]. The induction of c-di-GMP signaling pathway does not affect aerobic growth of *M*. *tuberculosis*, however, under hypoxia it plays an important role in bacterial survival and recovery from dormancy that is independent from the DosR regulon [65]. The negative regulation of c-di-GMP in *M. tuberculosis* virulence has been demonstrated in culture macrophages and in vivo [65]. Here we show that the absence of possible diguanylate cyclase *MAP_0949* gene severely attenuates MAP during macrophage infection and the phenotype can be recovered by complementing the mutant with the functional gene. Our results confirm an importance of MAP_0949 virulence factor in MAP survival within host macrophages.

The *MAP_2291* gene, knocked out in 18F6 mutant of MAP, encodes the globin protein and is found across *Mycobacterium avium* complex (MAC) and other subspecies of MAP including (MAP4, CLIJ623, Pt139). MAP_2291 protein is 86% homologue to *M. tuberculosis* oxygen-binding glbO. When expressed in *E.coli* and *M. smegmatis*, the globin protein highly stimulates the respiration activity and oxygen uptake, and increased association of glbO with membrane vesicles was also observed [66]. Membrane properties of globin has been validated to support the sequestration of oxygen for availability to intracellular *M. tuberculosis* under the hypoxic conditions [67]. The globin family of proteins have a unique and diverse sequence properties, but all share a function related to oxygen affinity and reactivity but also NO and CO [68]. *M. leprae* truncated hemoglobin O (trHbO) has a dual function of respiration and NO detoxification [69,70,71]. *M. tuberculosis* trHbO displays moderate NO-scavenging activity [72], signifying its involvement in both NO detoxification and aerobic respiration [73]. Due to the fact that MAP_2291 consists of the *M. tuberculosis* hemoglobin O like (TrHb2_Mt-trHbO-like_O) domain (E-value of 2.58007e-64), we can hypothesize that MAP_2291 globin protein may prevent the killing of intracellular MAP by protecting the pathogen from oxidative stress and microaerophilic (low oxygen) condition in macrophages.

The *MAP_3634* gene of deficient 15G2 mutant, also attenuated in intracellular growth, was identified as the hypothetical gene but to contain the IgD-like repeat domain of mycobacterial L,D-transpeptidases. The L,D-transpeptidase is responsible for the final polymerization steps involved in the formation of the glycan strands, and cross-linking peptide stems of the peptidoglycan cell wall in many bacteria [74]. L,D-transpeptidases have been identified in numerous bacteria including *Enterococcus faecium*, *Bacillus subtilis*, *Mycobacterium tuberculosis*, *Clostridium difficile*, and *E. coli* [75]. Inactivation of these transpeptidases has been indicated to be lethal in *M. tuberculosis* and *M. abscessus* [76,77,78], and carbapenems and amoxicillin antibiotics have been shown to inhibit L,D-transpeptidases of *M. tuberculosis* and ESKAPE pathogens [79]. Peptidoglycan modifications by L,D-transpeptidases has been demonstrated to control a secretion of bacterial virulence factors such as toxins [80], and deficiency in transpeptidases function can impair variety of processes including biofilm formation [81]. MAP lacking the *MAP_3634* gene most likely is attenuated in survival within macrophages due to defects effecting bacterial cell wall synthesis.

Altogether, our results indicate that MAP infection has an influence on the metabolic activity of the environmental protozoan host. Bacterial genes related to the metabolic shift in amoeba are also associated with the intracellular MAP burden in macrophages. Stimulation of metabolism appears to occur mainly paired with decreased viability of MAP, while a slower, less dramatic metabolic stimulation occurs at the same time that bacteria are replicating and increasing in number within the phagocyte. MAP grows at extremely slow rates and, in order to accomplish various phenotypic tastings, it requires several months of examination and validation. In this study, we confirm that amoeba can be used as a quick initial screening tool for discovery of genetic factors potentially contributing to virulence and pathogenicity of MAP. Several virulence targets identified in this study are currently being characterized.

## Figures and Tables

**Figure 1 microorganisms-08-01571-f001:**
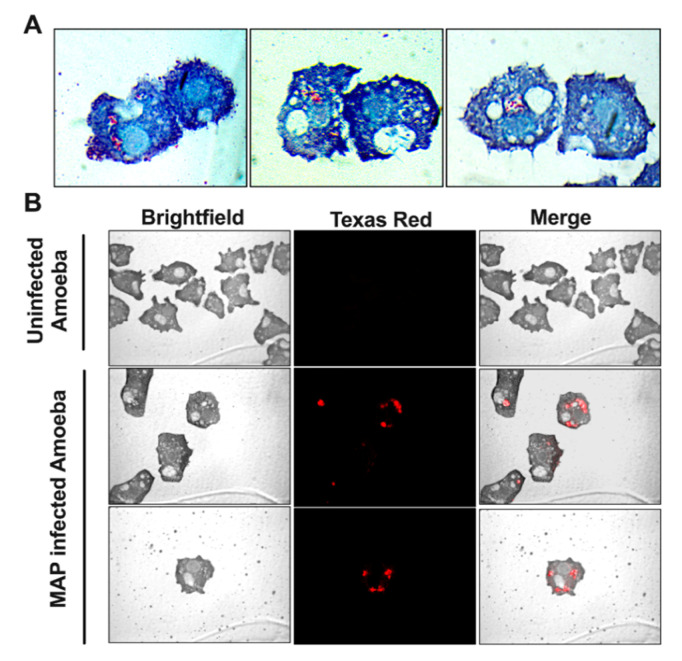
Amoeba readily phagocytize MAP during infection. Amoeba was incubated with either non-labeled (**A**) or carbol-fuchsin labeled (**B**) MAP at multiplicity of infection (MOI) of 10 and, after 1 h incubation, processed for Kinyoun acid-fast staining (**A**) or visualized under fluorescent microscopy (**B**). In micrographs, MAP is localized within the phagosome vacuoles and captured as red bacilli. Images are taken at 100× (**A**) and 40× (**B**) magnification and include uninfected amoeba as well.

**Figure 2 microorganisms-08-01571-f002:**
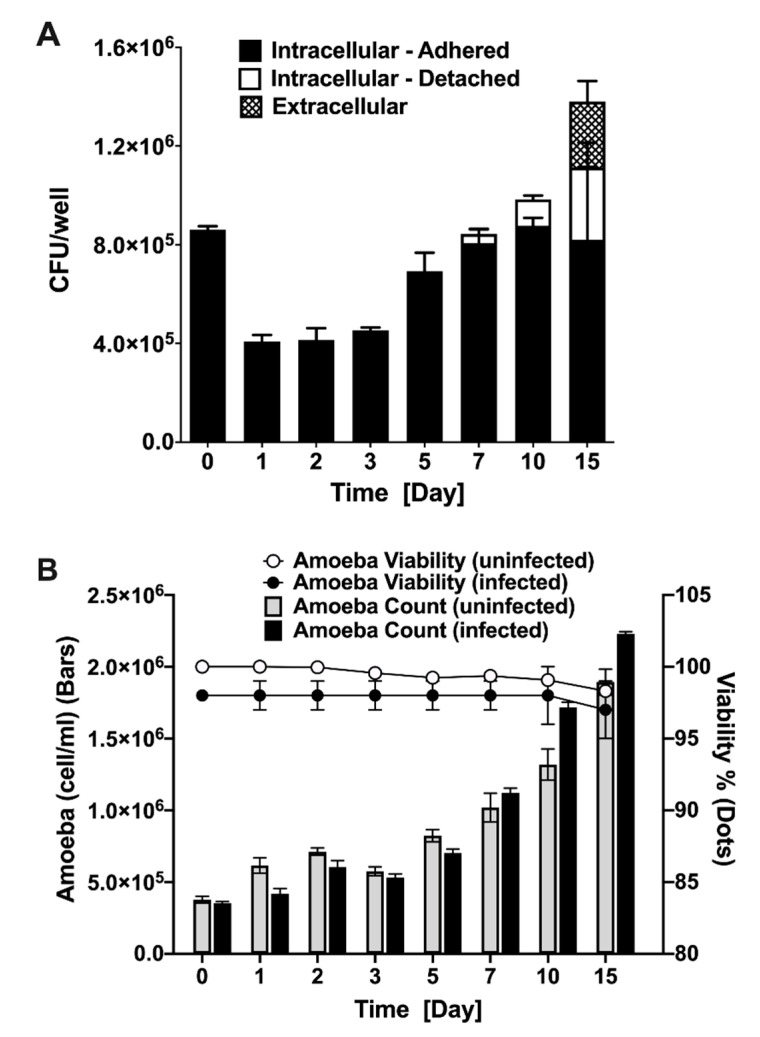
MAP and amoeba viability during infection. Amoeba was infected with MAP at MOI of 10 for 1 h followed by three washing steps and 2 h amikacin (200 µg/mL) treatment to remove extracellular MAP. (**A**) Both detached and attached amoeba were lysed and intracellular MAP was quantified with colony forming unit (CFU) counts. At each indicated time points, supernatants were also collected to quantify extracellular MAP. (**B**) The amoeba growth and viability in MAP-infected experimental and uninfected control groups were determined using trypan blue staining. The amoeba number was quantified over 15-day course of infection and data is presented as a bar chart. The dot chart shows the percentage of viable amoeba in both groups. Both MAP and amoeba data represent the mean ± SD of wells assayed in triplicate and is representative of two independent experiments.

**Figure 3 microorganisms-08-01571-f003:**
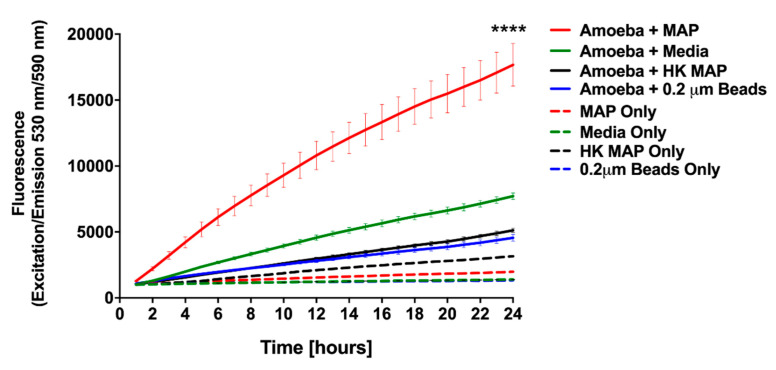
Stimulation of amoeba metabolism by MAP infection. Amoeba was infected with either live or heat-killed MAP (MOI 10). The 0.2 µm latex beads and no infection (PYG medium alone) served as controls. Plates were incubated for 1 h at 25 °C, and then extracellular bacteria, debris, or beads were removed by washing. Wells were replenished with fresh PYG medium containing 10% alamarBlue and fluorescence was measured at 530 nm/590 nm every hour for 24 h. Data represent the mean ± SD of two independent experiments conducted in triplicate. **** *p*-value < 0.0001 between live MAP-infected and all control groups at 24 h time point. HK, heat killed MAP.

**Figure 4 microorganisms-08-01571-f004:**
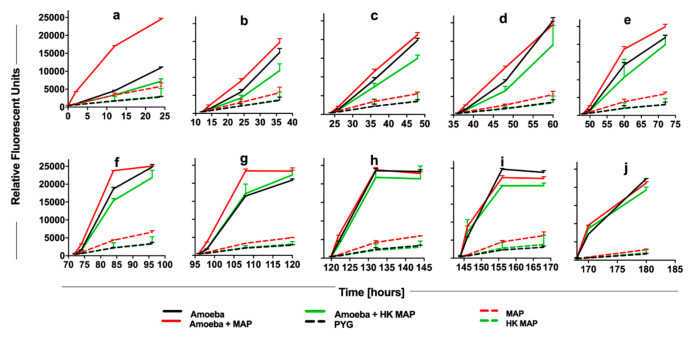
The long-term impact of MAP infection on amoeba metabolism. Amoeba was infected with either live or heat-killed MAP or added only Peptone-Yeast-Glucose (PYG) medium for 2 h. At 0 h (**a**), 12 h (**b**), 24 h (**c**), 36 h (**d**), 48 h (**e**), 72 h (**f**), 96 h (**g**), 120 h (**h**), 144 h (**i**), and 168 h (**j**) alamarBlue was added and fluorescent readings were recorded at 0, 2, 12, and 24 h after the addition of dye. Live and heat-killed bacterial controls in absence of amoeba were included to record the amoeba-independent reduction of dye. PYG media without bacterial and amoeba addition serves as a blank in the assay. The data represent the mean ± SD of three independent experiments each performed in triplicate. HK, heat-killed MAP.

**Figure 5 microorganisms-08-01571-f005:**
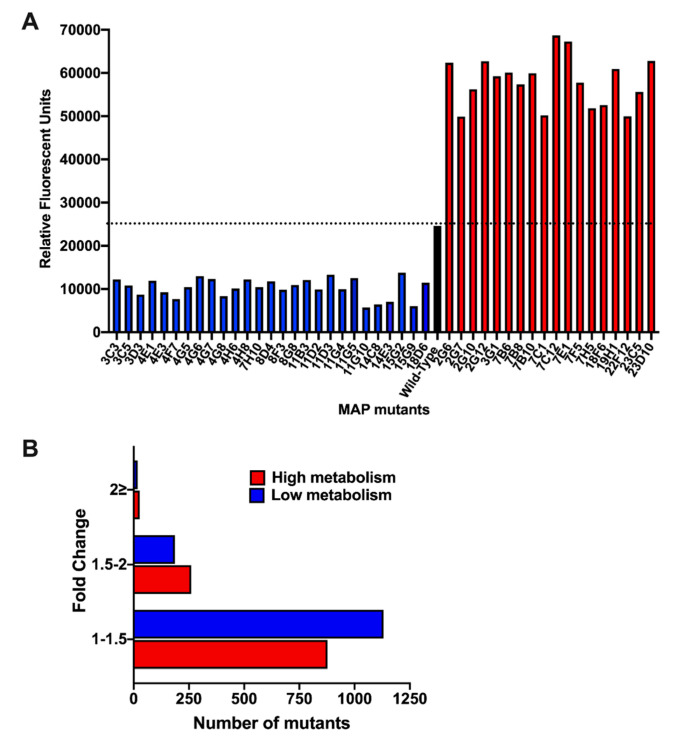
MAP transposon library screening for changes in amoeba metabolism. (**A**) *Acanthamoeba castellanii* monolayers seeded in 96-well plates were infected with the wild-type MAP or transposons mutants at MOI of 10:1 for 2 h. Extracellular bacteria were removed by washing cells with the Page’s amoeba saline, after which wells were replenished with 712 PYG media containing 10% (*v*/*v*) AlamarBlue dye. Fluorescent readings were recorded at 24 h of post-infection on a Tecan F200 plate reader. The dashed black line represents the metabolic activity of amoeba produced by the wild-type MAP infection. Mutants that displayed changes in amoeba metabolism by 2-times greater or less than the wild-type MAP are shown in red or blue bars, respectively. (**B**) The histogram demonstrates the distribution of MAP mutant library based on the fold-change in amoeba metabolism.

**Figure 6 microorganisms-08-01571-f006:**
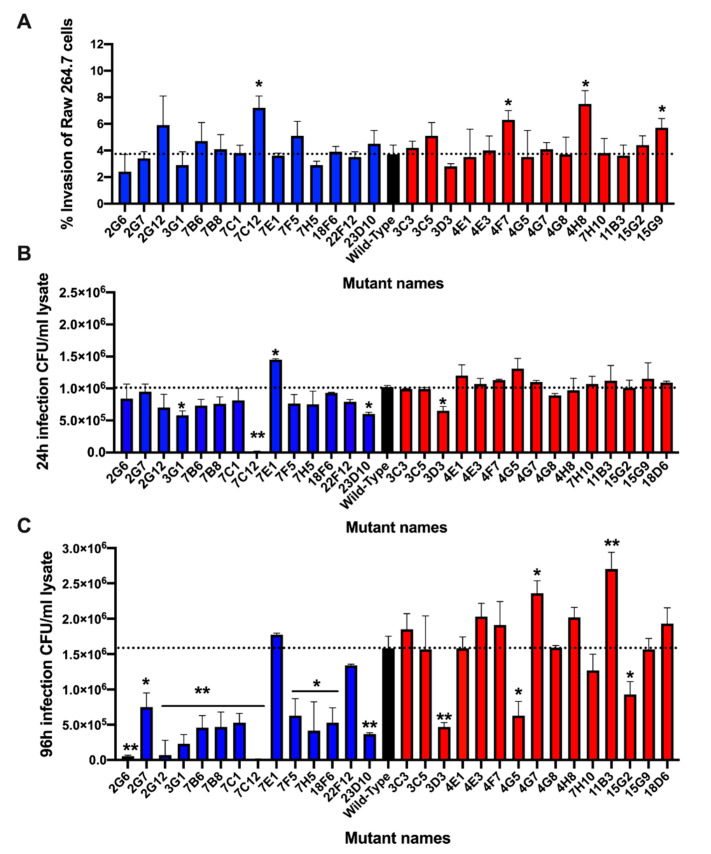
Verification of MAP mutants in Raw 264.7 macrophages. In order to validate if there was an association of amoeba metabolic variation with virulence mechanism of MAP, sequenced mutants were examined in Raw 264.7 macrophages for survival phenotype. Approximately, 10^6^ cells seeded in 24-well plates were infected with MAP or transposons mutants at MOI of 10 for 2 h. The extracellular bacteria were removed by washing steps and treating macrophages with 200 μg/mL amikacin. The intracellular bacteria were quantified at 2 h (baseline), 24 h and 96 h time points by plating serially diluted samples on 7H10 agar plates. (**A**) The percentage of MAP uptake by macrophages at 2 h was calculated from the original inoculum; (**B**) the intracellular bacterial growth at 24 h and (**C**) at 96 h are calculated from the baseline MAP that entered cells during 2 h infection. Data represent the mean ± SD of three independent experiments each completed in duplicate. *, *p* < 0.05 and **, *p* < 0.01 between the wild-type MAP and mutant infections at the corresponding time point.

**Figure 7 microorganisms-08-01571-f007:**
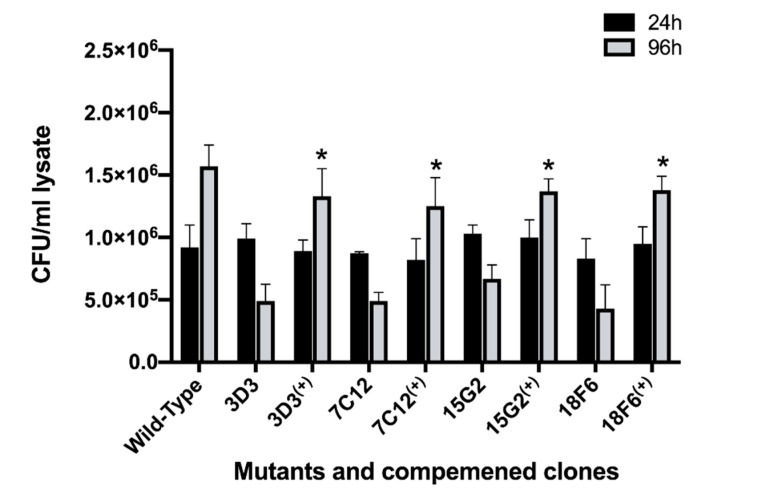
The growth rate of complemented MAP clones within macrophages. RAW 264.7 monolayers were infected with the wild type, the gene knockout mutants or complemented clones as described in materials and methods. The colony forming units of surviving intracellular bacteria were determined at one and four days post-infection. Data represent the mean ± SD of three independent experiments each completed in triplicate. *, *p* < 0.05 between the complemented clone and mutant at the corresponding time point.

**Table 1 microorganisms-08-01571-t001:** Primers used to complement *Mycobacterium avium* subsp. *paratuberculosis* (MAP) transposon mutants.

Gene Name Mutant	Sequence
*MAP_0949* 7C12	Forward (HindIII) 5′-TTTTTAAGCTTGTGCCACGCAGCCTGGAC-3′ Reverse (NotI) 5′-TTTTTGCGGCCGCCTAGATCGAGCCCATC-3′
*MAP_2291* 18F6	Forward (HindIII) 5′-TTTTTAAGCTTATGGATCAGGTGAGCTTC-3′ Reverse (NotI) 5′-TTTTTGCGGCCGCTCACAACGGAGAATTCAC-3′
*MAP_3634* 15G2	Forward (HindIII) 5′-TTTTTAAGCTTATGTCGGGCTGGACGAG-3′ Reverse (NotI) 5′-TTTTTGCGGCCGCCTAGTTCATCCAGTCG-3′
*MAP_3893c* 3D3	Forward (BamHI) 5′-TTTTTGGATCCATGGCCGAGCCGGA -3′ Reverse (EcoRI) 5′-TTTTTGAATTCTCAGAACGTGCTGGTGGG -3′

**Table 2 microorganisms-08-01571-t002:** List of genes interrupted in MAP mutants.

Mutant	Gene Name	Amoeba Metabolism	Attenuated in Macrophages	Function/Domains/Notes
15G9	*MAP_0800c*	Low	No	Helicase_C_3 domain-containing protein
2G7	*MAP_0122*	High	Yes	PE (proline-glutamate) family protein
4E3	*MAP_0294c/pca*	Low	No	Pyruvate carboxylase
18D6	*MAP_0338c*	Low	No	DUF772 domain-containing protein
3C5	*MAP_0824/aurF*	Low	No	Metalloenzyme P-aminobenzoate N-oxygenase
4F7	*MAP_0847*	Low	No	DUF4185 domain-containing protein
7C12	*MAP_0949*	High	Yes	Hypothetical protein with EAL domain/Diguanylate cyclase domain
7E1	*MAP_1024/cysM2*	High	No	Cystathionine beta-synthase
14C8	*MAP_1076*	Low	N/A	Hypothetical protein
2G12	*MAP_1133*	High	Yes	Methionyl-tRNA formyltransferase
7B10	*MAP_1221*	High	N/A	DNA-binding response regulator, OmpR family, contains REC and winged-helix (wHTH) domain
4H8	*MAP_1301/chaA*	Low	No	Ca^2+^/H^+^ antiporter
14E3	*MAP_1320c*	Low	No	Thiolase_N domain-containing protein; Lipid-transfer protein
23D10	*MAP_1423*	High	Yes	Major Facilitator Superfamily (MFS) domain-containing protein; Benzoate transport
4G7	*MAP_1450c*	Low	No	Flavoprotein CzcO associated with the cation diffusion facilitator CzcD
7B8	*MAP_1591*	High	Yes	Methylmuconolactone methyl-isomerase; EthD domain-containing protein
7B6	*MAP_1592*	High	Yes	Putative_PNPOx domain-containing protein
11D3	*MAP_1642*	Low	N/A	Lactamase_B domain-containing protein
11D2	*MAP_1824c*	Low	N/A	Integrase catalytic domain-containing protein
22F12	*MAP_1842c*	High	No	tRNA (adenine(58)-N(1))-methyltransferase TrmI
2G6	*MAP_2127*	High	Yes	Flavin-utilizing monoxygenases; Bac_luciferase domain-containing protein
3C3	*MAP_2228*	Low	No	Pimeloyl-CoA dehydrogenase
18F6	*MAP_2291/glbO*	High	Yes	Truncated hemoglobin, group 2 (O)
7H5	*MAP_2324c*	High	Yes	YdfJ uncharacterized membrane protein/MmpL
2G10	*MAP_2363c*	High	N/A	Acyl-CoA dehydrogenase related to the alkylation response protein AidB
11B3	*MAP_2535*	Low	No	Hypothetical protein
7F5	*MAP_2843c*	High	Yes	Hypothetical protein
3G1	*MAP_2973/llpW*	High	Yes	Penicillin binding protein transpeptidase domain-containing protein; Lipid transport
15G2	*MAP_3634*	Low	Yes	IgD-like repeat domain of mycobacterial L,D-transpeptidases
4G8	*MAP_3717c*	Low	No	Dipeptidyl aminopeptidase/acylaminoacyl peptidase domain
4H6	*MAP_3761c*	Low	N/A	Transmembrane sulfolipid-1 addressing protein SAP
4G5	*MAP_3832c*	Low	Yes	Molecular chaperone DnaK
3D3	*MAP_3893c/pknG*	Low	Yes	Serine/Threonine kinase
7C1	*MAP_3947/mmpl4_7*	High	Yes	Uncharacterized membrane protein mmpL
7H10	*MAP_4350c*	Low	No	50s ribosomal protein L34

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
