# Peer review of "Acanthamoeba castellanii* as a Screening Tool for *Mycobacterium avium* Subspecies *paratuberculosis* Virulence Factors with Relevance in Macrophage Infection"

_microorganisms, 2020, doi:10.3390/microorganisms8101571_

Round 1

Reviewer 1 Report

In this paper, Philips et al present their results regarding the effect of Mycobacterium avium subsp. paratuberculosis (MAP) in an in vitro model utilizing Acanthamoeba castellanii (amoeba). They also aim to explore metabolic consequences of the MAP-host interaction and establish a correlation between metabolic changes of this phagocytic host and MAP virulence. In addition, they were able to correlate the results generated in amoeba also in the higher eukaryotic setting by employing experiments in Raw 264.7 macrophage cell line. They conclude that the results generated within this study support the use of the amoeba model system as a quick initial screening tool for selection of virulence factors of extremely slow-grower MAP that is challenging to study and provide a map of several virulence factors of MAP mutants with biological relevance in mycobacterial survival and intracellular growth in phagocytic cells, uncovering new genetic targets potentially contributing to MAP pathogenicity. 

The paper is well written, easy to follow and presents the results in a comprehensive way. The authors' claims are fully supported by their results. A minor comment would be if it would be possible, albeit in a different paper, to present a head-to-head comparison of the global gene expression of the infected and non-infected amoeba cells early after infection, i.e. within the first 24 hours and after 8-10 days of infection. The results of such a study would be also of certain great interest.

Author Response

Reviewer 1.

In this paper, Philips et al present their results regarding the effect of Mycobacterium avium subsp. paratuberculosis (MAP) in an in vitro model utilizing Acanthamoeba castellanii (amoeba). They also aim to explore metabolic consequences of the MAP-host interaction and establish a correlation between metabolic changes of this phagocytic host and MAP virulence. In addition, they were able to correlate the results generated in amoeba also in the higher eukaryotic setting by employing experiments in Raw 264.7 macrophage cell line. They conclude that the results generated within this study support the use of the amoeba model system as a quick initial screening tool for selection of virulence factors of extremely slow-grower MAP that is challenging to study and provide a map of several virulence factors of MAP mutants with biological relevance in mycobacterial survival and intracellular growth in phagocytic cells, uncovering new genetic targets potentially contributing to MAP pathogenicity. 

The paper is well written, easy to follow and presents the results in a comprehensive way. The authors' claims are fully supported by their results. A minor comment would be if it would be possible, albeit in a different paper, to present a head-to-head comparison of the global gene expression of the infected and non-infected amoeba cells early after infection, i.e. within the first 24 hours and after 8-10 days of infection. The results of such a study would be also of certain great interest.

A: We agree with a reviewer that gene expression studies of MAP infected amoeba and comparative analysis together with macrophage gene expression will hold important information for understanding cellular pathways and some bacterial survival mechanisms that MAP acquired from amoeba through evolution. Thank you for the suggestion.

Reviewer 2 Report

Dear authors,

 The manuscript “Acanthamoeba castellanii as a screening tool for Mycobacterium avium subspecies paratuberculosis virulence factors with relevance in macrophage infection” is an interesting and well written manuscript. In this study, authors attempted to effect of the MAP infection on metabolic activity of amoeba. Authors elucidate relationship between metabolic changes of amoeba, MAP survival and intracellular burden in host macrophages.

Please see the comments below:

  1. There are several grammatical errors that authors need to pay attention to those errors

    2.    Fig. 1, it is important to discuss the morphology of intracellular MAP.

  1. In Fig. 2B, authors indicate that MAP infection do not have a detrimental effect on the overall viability of amoeba. What would be the possible reason for this observation? Also this sentence has some grammatical errors and please fix them.

  1. In 6B, authors measured readings at only 24 and 96 hrs. Are there any reasons to eliminate other time points between 24 and 96 hrs

  1. Based on the list of genes interrupted in MAP mutants, authors discuss the role of these genes in MAP virulence and pathogenesis. It is important to prepare a hypothetical model and connects the gene functions to see how they are interrelated.

Author Response

Reviewer 2.

The manuscript “Acanthamoeba castellanii as a screening tool for Mycobacterium avium subspecies paratuberculosis virulence factors with relevance in macrophage infection” is an interesting and well written manuscript. In this study, authors attempted to effect of the MAP infection on metabolic activity of amoeba. Authors elucidate relationship between metabolic changes of amoeba, MAP survival and intracellular burden in host macrophages.

Please see the comments below:

  1. There are several grammatical errors that authors need to pay attention to those errors.

A: We corrected grammatical errors and changes are mark in red throughout the text.

  1. 1, it is important to discuss the morphology of intracellular MAP.

A: We added the following sentence: In micrographs, MAP is localized within the phagosome vacuoles and captured as red bacilli. Lines 212-213.

  1. In Fig. 2B, authors indicate that MAP infection do not have a detrimental effect on the overall viability of amoeba. What would be the possible reason for this observation? Also this sentence has some grammatical errors and please fix them.

A: Amoeba have long been noted to interact and evolve co-existing strategies with mycobacterial species in the environment and becoming a significant part of mycobacterial evolution. MAP is frequently isolated from amoeba, and free-living amoeba has been frequently isolated from habitats common to mycobacteria supporting an “endosymbiotic” relationship between mycobacteria and free-living protozoan host. In fact, MAP does not kill macrophages either. They replicate within phagocytes and leave cells with a mechanism that we do not know yet.

We corrected the grammatical error. Lines 228-229    

  1. In 6B, authors measured readings at only 24 and 96 hrs. Are there any reasons to eliminate other time points between 24 and 96 hrs.

A: MAP is very slow growing bacteria and takes up to 48 hours to replicate within macrophages. At 24h time point, we record survival bacteria after infection because not all bacteria that invade macrophages survive after infection, and then at 96h to record detectable differences in MAP intracellular growth.

  1. Based on the list of genes interrupted in MAP mutants, authors discuss the role of these genes in MAP virulence and pathogenesis. It is important to prepare a hypothetical model and connects the gene functions to see how they are interrelated.

A: The reviewer’s suggestion for creating a model is valid but we believe that selected genes have independent contribution to MAP virulence and pathogenicity. For example:  MAP_0949 is an uncharacterized protein that contains the EAL domain/diguanylate cyclase domain lic di-GMP, and due to the fact that it has a secreted signal peptide, probably it is secreted from bacteria. We aren’t sure when this protein is secreted - during invasion after infection(?), but it possibly triggers signaling to promote anaerobic growth within macrophages. The MAP_3893c is another uncharacterized protein but has significant similarity to the serine/threonine kinase (PknG) protein in M. tuberculosis where it was demonstrated to block phagosome acidification. The truncated hemoglobin group 2 (O) protein MAP_2291 (glbO), possibly can prevent intracellular MAP killing by protecting the pathogen from oxidative stress, while the MAP_3634 hypothetical gene, which contains IgD-like repeat domain of L,D-transpeptidases, possibly can effect mycobacterial cell wall integrity.

Our discussion and hypothesis are based on findings gathered from published literature on mycobacterial or other bacterial species and, unless experimentally proven, the model will be based on too much speculation.

The aim of this study was to develop the fast screening tool for selection of MAP virulence factors and these genes need be further studied to define their association and mechanism in MAP pathogenesis.